# A Piezoelectric Sensor Signal Analysis Method for Identifying Persons Groups

**DOI:** 10.3390/s19030733

**Published:** 2019-02-12

**Authors:** Hitoshi Ueno

**Affiliations:** Tokyo-Ueno Campus, Daiichi Institute of Technology, Tokyo 110-0005, Japan; h.ueno@ueno.daiichi-koudai.ac.jp

**Keywords:** watching system, piezo-electric sensor, bio-signal, cardiac signal, personal identification

## Abstract

The is an increasing number of elderly single-person households causing lonely deaths and it is a social problem. We study a watching system for elderly families by laying the piezoelectric sensors inside the house. There are few privacy issues of this system because piezoelectric sensor detects only a person’s vibration signal. Furthermore, it has a benefit of sensing the ability for a bio-signal including the respiration cycle and cardiac cycle. We propose a method of identifying the person who is on the sensor by analyzing the frequency spectrum of the bio-signal. Multiple peaks of harmonics originating from the heartbeat appear in the graph of the frequency spectrum. We propose a method to identify people by using the peak shape as a discrimination criterion.

## 1. Introduction

Along with the progress toward an aging society, the number of elderly people living alone grows dramatically, and solitary death became a social problem. As a solution to this, watching facilities that make it possible to monitor elderly homes from remote supporters have been proposed. In previous studies, we have proposed a senior-citizen-watching system that monitors elderly residents without wearing sensors on themselves by using sheet-type piezoelectric sensors [1,2]. Since non-restricted sensors are used for subjects in this system, this has a merit that biological signals can be acquired in the case where abnormality occurs, even though elderly people do not want to wear wearable sensors. There have been other reports on non-restricted sensors used for watching the elderly living alone so far, but these have not been confirmed to be effective in the monitoring of multiple subjects [3,4,5,6].

The problem is living with a few families, but not necessarily living alone. There are cases where elder people live with couples or stay with visitors. Therefore, it is desirable that the remote monitoring system has a function to distinguish individuals in which abnormality has occurred.

We have found that the frequency spectrum of the signal obtained by the piezoelectric sensor shows a different pattern for each individual. By using this principle, even when there are multiple residents in the elderly person’s home, it becomes possible to determine who has collapsed from that signal and the watching system becomes more useful.

From the viewpoint of frequency spectrum analysis of heart beat vibration, it is similar to a heart rate variability (HRV) analysis, which analyzes the situation of autonomic nerves. However, in HRV analysis, we analyze the frequency band from 0.0033 Hz to 0.4 Hz [7], whereas in this study we analyze signals with high frequencies, such as 0.3 Hz to 15 Hz, meaning that it is a completely different analysis method.

## 2. Study of Elderly Watching System

For the purpose of preventing solitary death in those living alone or living with elderly couple, we study a senior citizen watching system to monitor the health condition by laying seat-type piezoelectric sensors in the house (Figure 1).

Recently, there are two types of watching systems that are selling products or technologies under study. One is a lifestyle habits monitoring feature, and another is a monitoring feature of bio-signals using a wearable sensor. The former example are a feature that uses an electric pot connected to the Internet or a monitoring the frequency of use of the toilet by attaching a sensor to the door of the toilet. The latter includes wearing a wrist watch type bio-signal sensor or a simple electrocardiogram.

In the former method, even if a health abnormality occurs in the elderly to be monitored, there is a problem that it takes time on a day-by-day basis to detect a health problem, and further, it is impossible to acquire a biological signal at the time of abnormality detection. In the latter case, the elderly must always wear a sensor, and there is the problem that some people do not want to wear sensors.

In the system proposed by us, since sensors are installed in houses, there is no burden for the elderly themselves to wear sensors. In addition, in the case of collapsing in the house, by analyzing the signal of the seat type piezoelectric sensor, it is possible to acquire biological signals related to cardiac state and respiration, such that there is a feature that it is possible to grasp a certain degree of health condition at the time of the health problem. That is, it is possible to realize an elderly watching system capable of detecting a problem with no constraint and acquiring a bio-signal at that time.

In this method, although it is possible to acquire a bio-signal of a person on the sensor, there is a problem that it is not clear who the person is. Therefore, in this research, we investigate whether there is the possibility of personal identification to some extent from the measured bio-signal.

## 3. Bio-Signal from Sheet-Type Piezoelectric Sensor

### 3.1. Measurement of the Sensor

The sheet-type piezoelectric sensor used in this research has a shape with an external dimension of 30 cm × 30 cm using PVDF (polyvinylidene difluoride). The sensor was placed on the chair and the person sat on it. They were then measured for 3 min, and the acquired vibration waveform data obtained by the piezoelectric sensor was stored in the personal computer (PC) for analysis (Figure 2).

It is known that the cardiac cycle obtained by a piezoelectric sensor is equivalent to that obtained by an electrocardiograph (ECG) [8,9]. By using different frequency filters, the PC used for analysis separately displayed this vibration signal as a respiratory cycle waveform and a cardiac cycle waveform.

In our research on this frequency filtering method, we found that the individual difference was large in the vibration component derived from the vibration of the heartbeat. The heartbeat is usually considered to be a wave with a frequency around 1 Hz, but some of the vibration components derived from the heartbeat detected by this sensor contained almost no frequency components around 1 Hz. In the case of such a subject, a higher harmonic component is detected as a waveform that was AM (Amplitude Modulation)-modulated at a frequency around 1 Hz. Therefore, there were subjects who simply could not be separated using the cardiac cycle signal by simply applying a frequency filter around 1 Hz.

This phenomenon is a problem from the viewpoint of extracting the cardiac signal by applying a unified algorithm to all subjects. However, on the other hand, if the vibration component derived from the cardiac is analyzed in detail, it means that there is a possibility that the subject on the sensor can be identified by the phenomenon. In this study, we analyze the frequency spectrum of the heartbeat component in detail. 

### 3.2. Signal Range to Extract a Bio-Signal

The original signal obtained from the piezoelectric sensor contained unstable components, such as body movement components and external noise, in addition to the biological signal component (Figure 3). Therefore, we had to extract a stable signal part (10 s) containing only the biological signal component and use it for analysis. This was the minimum necessary time for detecting “heart rate variability” to be described later, which was also the time to obtain stable signal data from most subjects.

The stable part of the original signal waveform contained not only the signal component derived from the heart beat but also a long period signal derived from respiration and a short period signal as a noise component. Therefore, we cut signals below 0.6 Hz and higher than 15 Hz and calculated the frequency spectrum for that signal.

## 4. Experimental Method

### 4.1. Method of Collecting Subject Data

By analyzing the sensor signal waveform, we presumed that it was possible to obtain patterns of different frequency distributions from different subjects, and similar patterns could be obtained even if we compared data from the same subjects acquired at different points in time. Therefore, we carried out two kinds of experiments.

The first was an experiment to collect different multi-person data, and we got data from 24 subjects. The other is an experiment to acquire data from one person three times on different days, and data was acquired from four subjects.

### 4.2. Data Collected from Different Multiple Subjects

Data was acquired from 24 subjects, and here the graphs of the signal waveform and the frequency spectrum for the data of all subjects are shown in Appendix A. The labels at the left end of each graph indicate the number of the subject: M01 to M05 are male and F01 to F03 are female. We chose a stable signal waveform period of 10 s from the 3-min measurement data of each subject and used it as the original signal. On the other hand, the calculation result of the frequency spectrum is the graph on the right side of each subject. The frequency spectrum is indicated by a normalized value so that the integral value of all sections became 1.

### 4.3. Data Collected from the Same Subject on Different Days

We obtained data from four subjects, where the data for four people is shown in Appendix B. These are graphs of signal waveforms measured three times on different days with the calculation results from the frequency spectrum. The left side of the graph of each measurement day is the part where the stable 10-s period of the original signal was selected, and the right side is the calculation result of the frequency spectrum of the signal.

## 5. Individual Identification Based on Spectrum

### 5.1. Detection Method of a Different Person

If the frequency spectrum of the signal obtained from the subject sitting on the sensor is different for each subject, it is possible to know from which signal frequency is emitted from the body of the subject based on the frequency spectrum. In order to identify individuals more accurately, it is desirable that the characteristics of the frequency spectrum be as different as possible from person to person.

Therefore, we performed an experiment sitting on the piezoelectric sensor for 24 subjects, and obtained a graph of the frequency spectrum shown in Appendix A. We assume that different individual characteristics can be determined from the following viewpoints.
the number of obvious peak shapes;sharpness of peak shape of peak frequency.

In order to represent these quantitatively, we propose the following calculation method.

Number of Peak Shapes
Using the frequency of the maximum power value as a reference, search for a peak frequency having power more than 1/2 thereof and count the number of peak shapes.Similarly count the number of peak shapes with more than 1/4 power.Define the average of the above (a) and (b) as the number of peak shapes.

Sharpness
Calculate the area of the section convex upward at each of the peak frequencies found based on the 1/2 standard of above (a).Similarly, calculate the area of the section convex upward with respect to each of the peak frequencies found based on the 1/4 standard of above (b).Add the area of both and divide by the number of peak shapes and calculate the reciprocal number. This is defined as the sharpness of the peak shape. The sharper the shape, the larger the value.

According to this definition, the distribution diagram plotting 24 sets of data for only one day (others) and data for four persons for 3 days (M05, M17, F09, M07) is shown in Figure 4. The horizontal axis represents the number of peaks, and the vertical axis represents the peak shape. From this distribution map, it can be seen whether the same subjects are distributed in a relatively close region, and whether different subjects are distributed in a wide range.

The reason why there is a strong correlation between peak number and sharpness is due to dividing the area of the peak by the number of peaks, and there is no significant principle of strong correlation. The remarkable item from the viewpoint of person’s differences is the magnitude of the variation with respect to the regression line. The larger the variance, the better we can use this as personal identification data.

### 5.2. Consistency of the Same Person

From the examination in the previous section, it was found that the frequency spectrum obtained from the signal waveform of the piezoelectric sensor differs for each subject. Next, we will consider whether the same person always has a similar frequency spectrum. The object of measurement in this report is a biological signal, and there is a possibility that a considerably different frequency spectrum appears depending on physical condition and time elapsed.

Appendix B is a graph in which measurement data of three different days are recorded for three subjects. Each subject contributed data three times within a period of 2 months to 6 months. Although the waveform of the original signal (left) seemed to be a seemingly different pattern depending on the data sampling time, even for the same subject, its frequency spectrum clearly had a similar shape. It was also clear that the frequency spectra between the different subjects M17 and M05 were not similar.

Therefore, for the same person, the characteristic of its frequency spectrum was considered to maintain considerable similarity, even if re-measurement was done after a long period of time such as several months later.

### 5.3. Application to Personal Identification

According to the examination in the previous section, it was found that characteristics of different frequency spectra can be obtained if subjects are different, and almost similar characteristics can be obtained if the subjects are the same. That is, if the characteristics of the frequency spectrum are registered in advance on an individual basis, when new signal data of the piezoelectric sensor is obtained, it is possible to specify who the data belongs to.

## 6. Consideration

It is considered that the biological signal obtained from the sheet-type piezoelectric sensor is detecting the vibration propagated from arteries in the body. Therefore, the cardiac cycle obtained by the measurement of this study is the heartbeat cycle obtained from the real-time blood pressure change in the artery. There is “fluctuation” due to respiratory arrhythmia in the cardiac cycle, and the time width of fluctuation is known that young people’s one is larger than elder’s. Specifically, it means that the time interval of the beating of the heart is not constant, which is a phenomenon with a variation of about 100 ms. For example, it indicates that the heartbeat time interval of the subject who beats 60 times per minute fluctuates in the range of about 1 s ± 100 ms at the maximum. In terms of frequency, if the fundamental frequency is assumed to be 1 Hz, it means that there is a variation of 1 Hz ± 0.1 Hz.

It is known that there is another factor influencing the blood pressure waveform. It is a reflected wave of the pressure generated in the blood vessel when the heart delivers blood to the artery. Since the extent of the pressure reflex varies depending on the hardness of the blood vessels of the individual and the viscosity of the blood, individual differences also appear in the blood pressure waveform. The distortion of the blood pressure waveform appears as an increase in high frequency components when observing the frequency spectrum, so it is considered that the signals sampled using the piezoelectric sensor appeared as the difference in shape of the frequency spectrum.

Figure 5 shows the relationship between the frequency spectral features and the age of the subjects. Heart rate fluctuation is said to have a high correlation with age. Therefore, it was assumed that the spectrum features are also influenced by age, but the scale used this time had little relevance to age.

From now on, by studying more data, we need to consider a more effective personal identification scale.

## 7. Conclusions

We are developing a watching system that spreads sheet-type piezoelectric sensors inside the house for the purpose of making it possible for the elderly people to be constantly monitored from remote support organizations via the Internet. Basically, a care person observes the biological signal from a remote place and assumes a use case for judging whether there is a possibility of an emergency situation requiring medical treatment when an elderly person living alone falls ill. However, even in the house where there are multiple people in the future, we are planning to extend it to a system for judging who is abnormal if there is abnormality in the biological signal.

The purpose of this research is to develop a method to enable individual identification in a certain range from signal data obtained using a sheet-type piezoelectric sensor for the purpose of using in a family environment of plural people.

Different test subjects showed different frequency spectra according to the subject data and the frequency spectrum analysis method reported this time, and it was found that similar frequency spectrum can be obtained even if the same person provides data on different days. This means that individual identification is possible using signal data obtained from the piezoelectric sensor.

However, the discrimination scale proposed this time is considered to be a relatively small individual difference and provides a low level of personal identification. In order to realize individual identification with high efficiency and high precision, we will study improvement of the scale in the future.

## Figures and Tables

**Figure 1 sensors-19-00733-f001:**
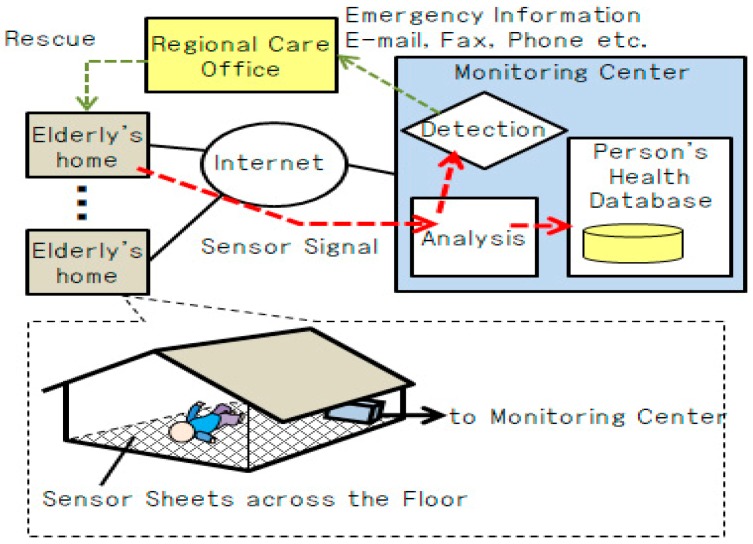
Configuration of elderly home monitoring system.

**Figure 2 sensors-19-00733-f002:**
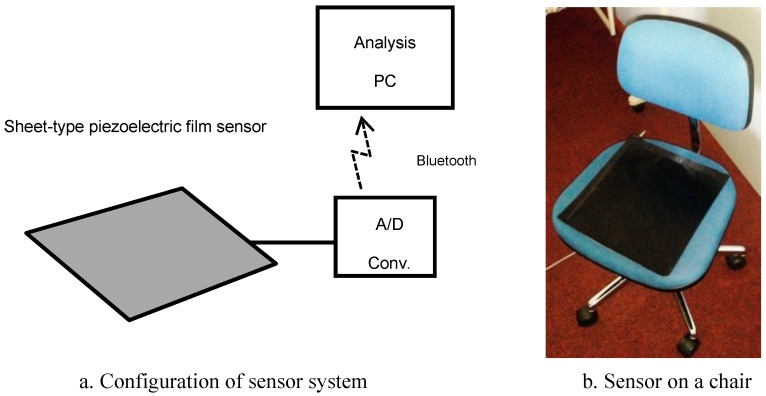
Measurement equipment.

**Figure 3 sensors-19-00733-f003:**
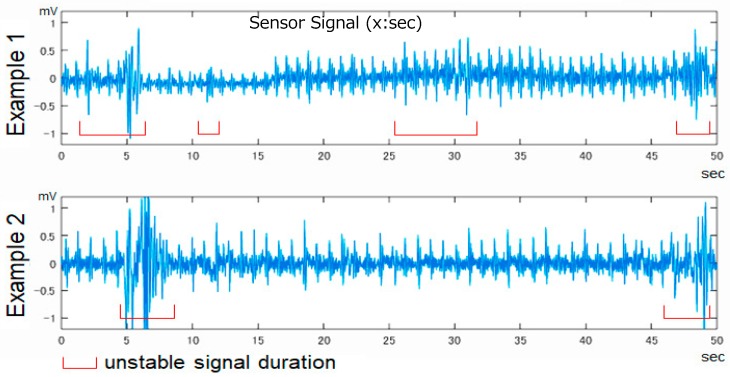
Original signal from the sensor.

**Figure 4 sensors-19-00733-f004:**
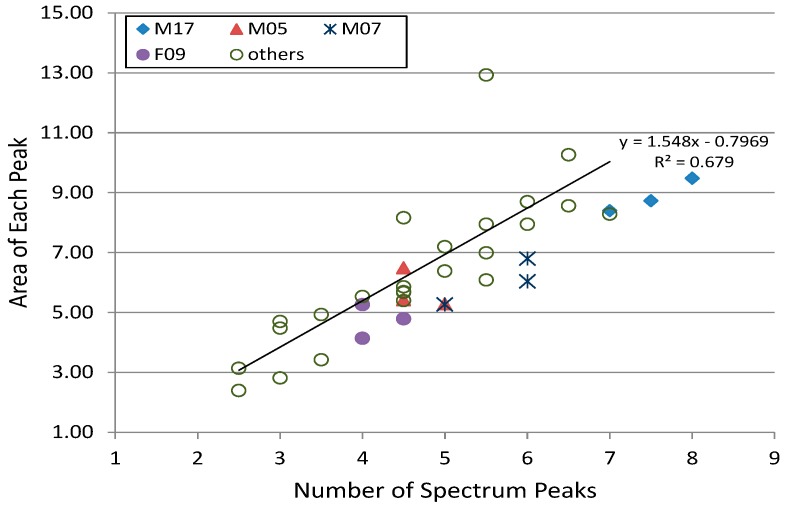
Relationship between number of peaks and sharpness.

**Figure 5 sensors-19-00733-f005:**
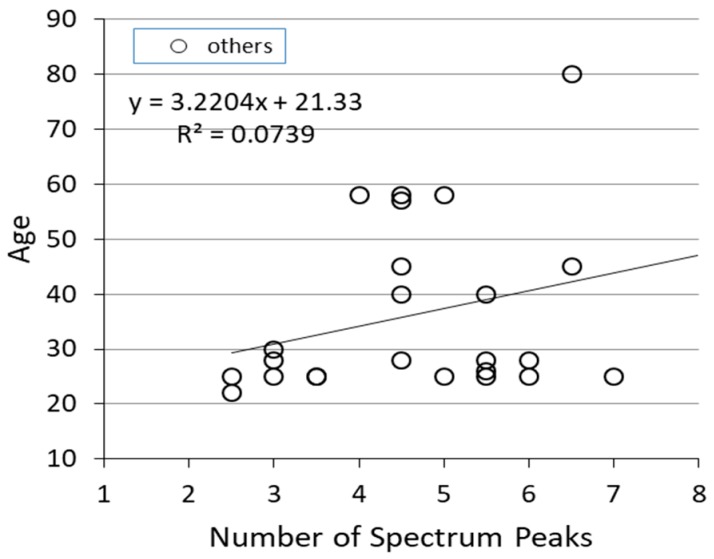
Relationship between number of peaks and age.

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
