# Peer review of "A Piezoelectric Sensor Signal Analysis Method for Identifying Persons Groups"

_sensors, 2019, doi:10.3390/s19030733_

Reviewer 1 Report

In this paper, the author reported a method to identify people by using piezoelectric sensors configured on the chair. This work aims at monitoring senior people who live alone to prevent solitary death. This work lacks technical innovation and the results are not convincing. The impact of this work is questionable. This work may be eventually published on Sensors after major revision.

(1)    If the purpose is to monitor elderly people living alone, why do you bother to identify different individuals through the sensor signal, since there is only one person in the room?

(2)    The English of this manuscript need to be improved.

(3)    It is understandable that the subjects need to use the toilet frequently every day. But why they must use chair in their daily life?

(4)    What's the mechanism behind the detection of respiratory cycle and cardiac cycle through the piezoelectric sensor on the chair?

(5)    Fig 3 has wrong caption. there is no unit in the graph.

(6)    The heartbeat of the respiratory frequency of the same subject vary at different time of the day. How to deal with this?

(7)    It lacks rationale behind the data processing algorithms.

(8)    There is lack of clear definition for the "peak shape" and there is no criteria for the "obvious peak shape"

(9)    What does "others" mean in Fig. 4?

(10) The data doesn't suggest that personal identification can be achieved through this method, since the variation for the same subject is so large.

(11) If the purpose is to monitor elders, why the ages of most of the subjects shown in Fig 5 are less than 50 years?

(12) There is no convincing data to shown that the breath frequency and the heart rate of the subjects can be detected by the piezoelectric sensor on the chair. This makes all the data analysis questionable.

Author Response

Thank you for your comments.

I correct my manuscript as follows, especially Introduction.

(1) My research target includes elder people living with couple or staying visitors.

(2) I will improve English later.

(3) This research is for evaluation of characteristics of one sensor sheet. In future target, I will lay many sensor sheets in whole of house. 

(4) The sensor detects vibration of the buttock artery.

(5) I revised the graph.

(6) This paper shows that the spectra of different days are similar. I think that the spectra are similar to each other at different times on the same day than for different days.

(7) Although the theoretical basis can not be proved, it showed that the spectrum of the vascular oscillation waveform is different depending on the person, as a result of actual measurement. This is presumed to be caused by the fact that the physical factors such as the shape of the blood vessel and the stickiness of the blood are different among individuals. It is empirically common sense to be able to distinguish people by voices which are vibrations of the vocal cords, but similarly, it is pointed out that there is a possibility that people can be identified by vibration of blood vessels.

(8) I explained about the calculation method in section 5.1.

(9) In Fig.4, 'others' means different subject with each other. In this graph, subjects, Mxx and Fxx are measured three times each in different day, but 'others' are measured only once.

(10) We can not show that individuals can be identified separately. However, since the difference in individuals is within a certain range, we can show that we can distinguish them into several groups. In this paper I think that it is important to show that it can be identified with a certain accuracy.

(11) In this paper, I reveal that the identification of individuals is possible with a certain precision with the spectrum of piezoelectric sensor signals. How to use this principle is widespread. My research is triggered by the elderly watching system, but the scope of this paper is not limited to the elderly. Therefore, the subjects also have many young people.

(12) It is clear from reference [2] that heartbeat signals and respiratory signals can be acquired with a piezoelectric sensor.

Thank you.

Reviewer 2 Report

The scientific impact of this manuscript is really promising, even as a statistic method in piezoelectric sensor results’. Aim of the proposed method, is to identify individuals in a place with many people, based in the bio-signal’s frequency spectrum analysis. Some things however must be addressed:

From the paper is also unclear the health      status of the subjects, i.e. if there are healthy or they have some heart      disease problems that may affect the heart rate and therefore the results.      

Also, in the Consideration part of the      paper, the author includes things such as respiratory arrhythmia and blood      pressure, which will be good if they have a theoretical solution as a      future problem to be solved.

You should provide more evidence regarding      the related work. Furthermore, references in relatively works are needed.

However regarding the experimental and technical tasks, they are very sufficient in description and methodology. The major added value of the paper is the originality and the feasibility of the precedent statistical methods in the signal analysis. Authors has used smartly and elegantly existing methods, aiming to obtain quantitatively scientific results.

The manuscript of the paper can be published as soon as the comments above will be addressed.

Author Response

Thank you for your comments.

I revised my manuscript as follows.

(1) I explain back ground more clearly.

(2) I attach new reference [7], it describe another spectrum analysis of heart beat signal.

Thank you.

Round  2

Reviewer 1 Report

Changes have been made according to reviewers suggestions. Can be published as is.